# Carbon-Based Quantum Dots for Electrochemical Detection of Monoamine Neurotransmitters—Review

**DOI:** 10.3390/bios10110162

**Published:** 2020-10-31

**Authors:** Saheed E. Elugoke, Abolanle S. Adekunle, Omolola E. Fayemi, Bhekie B. Mamba, El-Sayed M. Sherif, Eno E. Ebenso

**Affiliations:** 1Material Science Innovation and Modelling (MaSIM) Research Focus Area, Faculty of Natural and Agricultural Sciences, North-West University (Mafikeng Campus), Mmabatho 2735, South Africa; elugokesaheed@gmail.com (S.E.E.); sadekpreto@gmail.com (A.S.A.); omololaesther12@gmail.com (O.E.F.); 2Department of Chemistry, School of Physical and Chemical Sciences, Faculty of Natural and Agricultural Sciences, North-West University (Mafikeng Campus), Mmabatho 2735, South Africa; 3Department of Chemistry, Obafemi Awolowo University, Ile-Ife 220005, Nigeria; 4Nanotechnology and Water Sustainability Research Unit, College of Science, Engineering and Technology, University of South Africa, Johannesburg 1709, South Africa; Mambabb@unisa.ac.za; 5Center of Excellence for Research in Engineering Materials (CEREM), King Saud University, P.O. Box 800, Al-Riyadh 11421, Saudi Arabia; emsherif@gmail.com; 6Electrochemistry and Corrosion Laboratory, Department of Physical Chemistry, National Research Centre, Cairo 12622, Egypt; 7Department of Chemistry, College of Science, Engineering and Technology, University of South Africa, Roodepoort 1710, South Africa

**Keywords:** carbon quantum dots, graphene quantum dots, neurotransmitters, electrochemical sensors

## Abstract

Imbalance in the levels of monoamine neurotransmitters have manifested in severe health issues. Electrochemical sensors have been designed for their determination, with good sensitivity recorded. Carbon-based quantum dots have proven to be an important component of electrochemical sensors due to their high conductivity, low cytotoxicity and opto-electronic properties. The quest for more sensitive electrodes with cheaper materials led to the development of electrochemical sensors based on carbon-based quantum dots for the detection of neurotransmitters. The importance of monoamine neurotransmitters (NTs) and the good electrocatalytic activity of carbon and graphene quantum dots (CQDs and GQDs) make the review of the efforts made in the design of such sensors for monoamine NTs of huge necessity. The differences and the similarities between these two quantum dots are highlighted prior to a discussion of their application in electrochemical sensors over the last ten years. Compared to other monoamine NTs, dopamine (DA) was the most studied with GQDs and CQD-based electrochemical sensors.

## 1. Introduction

Every human action (voluntary or the converse) has a connection with the nervous system. These actions, ranging from very essential ones such as learning, memory, sleeping, consciousness, heart rate and muscle contraction to seemingly inconsequential ones like relaxation, are coordinated by the central and the peripheral nervous system [1]. The nervous system, just like every other system sends and receives information via specific channels. These channels, which form the fundamental unit of the entire nervous system are termed nerve cells (or neurons). From a certain neuron to the other, information is transferred as either electrical or chemical signals [2]. Notably, each nerve cell comprises of the axon, through which signals are sent, dendrites, which receive such signals, and the cell body (or soma), which keeps the entire cell alive [3]. The chemical signals are more often given special scientific considerations because they are much more easily controlled than the electrical signals. These signals are sent when chemical messengers are transported across a junction between two neurons, otherwise called the synapse [4]. These chemical messengers are called neurotransmitters [5].

Neurotransmitters (NTs) have to be at the appropriate concentration in the nervous system for proper coordination of the aforementioned human actions. It has been scientifically proven that depletion, upshoot or imbalance of these very important chemical messengers have been implicated several physiological and psychological disorders such as schizophrenia, epilepsy, addiction, Parkinson’s disease, and Tourette’s syndrome, among others [6,7,8]. Therefore, it is understandable that scientists have gone the extra mile to devise a means of detecting the level of these chemicals in the human extracellular fluids as a means of diagnosing patients with one physiological disorder or another [9,10,11,12,13]. Essentially, neurotransmitters have been classified based on their behaviour between two neurons as excitatory or inhibitory and according to their chemical composition. Specifically, those with the single amino functional group such as Serotonin, Dopamine, Epinephrine and Norepinephrine are classified as monoamine neurotransmitters [14].

Monoamine neurotransmitters (Figure 1) can be considered an extremely important class of NTs due to their importance in the coordination of the affairs of the human body system. Dopamine (DA), an inhibitory and excitatory neurotransmitter (NT) is responsible for motor activity, behaviour, mood and learning [15] and as such, an upset in its balance in the human system has been implicated in disorders such as Parkinson’s disease, dementia, epilepsy, schizophrenia and melancholia [16]. Epinephrine (EP), an excitatory NT helps in regulating alertness, cognition, metabolism as well as mental focus [17]. 

Consequently, an upshot in its concentration beyond the optimum level in the nervous system has been found to cause hypoglycaemia, and its imbalance has been found to be manifested in Parkinson’s disease as well [18]. Norepinephrine (NE), another excitatory neurotransmitter, is vital for metabolism, heart rate, and attention [19,20]. It is formed from dopamine and turns out to be important for forming memories, as well. Upsets in the NE balance of the body cause high blood pressure and anxiety [21,22]. Serotonin (SE), an inhibitory neurotransmitter is a special kind of NT that can also act as a neuromodulator [23], just like DA. SE is responsible for appetite, mood and sleep regulation [24,25]. Irregularities in its optimization in the nervous system have been seen to manifest in illnesses such as liver regeneration, depression, anxiety and psychosis [25,26]. Cumulatively, these classes of NTs have a great impact on the smooth running of the nervous system, and by extension, the entire body system. Therefore, their quantification in extracellular fluid in vivo or in vitro [27,28,29] could contribute to the easy diagnosis of health conditions associated with the imbalance in the level of any or all of these NTs in the human body system. 

Compared to other means of determination of these NTs, like electrochemiluminescence [30], spectroscopy [31] and chromatography [32], electrochemical techniques stand out as a result of their simplicity, ease of handling and cost-effectiveness [15]. A series of materials such as metal nanoparticles [33], metal oxide nanoparticles [15], carbon nanotubes (CNTs) [28], graphene [4], graphene oxide [34], conducting polymers [35], nafion [36] and imprinted polymers [37] have been used for the design of electrochemical sensors with a reasonably low limit of detection and sensitivity. In cases of detection of these NTs in the presence of interfering substances like ascorbic acid, tryptophan, uric acid and acetaminophen [35], some of these sensors have given good peak resolutions. However, the field of nanoscience, in a quest for highly conducting smaller materials with tunable electronic and optical properties, came up with a material that perfectly fits this description, and called it the quantum dot (QD).

Quantum dots, loosely called ‘artificial atoms’, are nanocrystals with extremely small sizes. The nomenclature ‘artificial atoms’ is the product of their possession of quantum effects just like conventional atoms, coupled with the fact that they are tailor-made materials [38]. Interestingly, their size and shape have a direct link with their optical and electronic properties, and this favourably distinguishes them from conventional atoms. Specifically, reducing their sizes further increases their bandgap and consequently, the emerging fluorescence post-emission of UV-light can be considered a function of their size [39,40]. Generally, quantum dots are either inorganic semiconducting materials or simply organic. Inorganic semiconducting QDs are structurally composed of an inner core, outer shell, stabilizing layer [41], ligands and the supporting linkers (Figure 2). The cores, such as ZnS, ZnO, PbS, ZnSe, CdS, etc. [42,43,44], can be either of the same composition as the protective shell (ZnS, ZnO, PbS, ZnSe, CdS) or different (PbTe, CdTe) [45,46,47]. This is not to say, however, that such semiconducting QDs cannot exist independently [48]. The relatively high toxicity of the inorganic semiconducting QDs have limited their application in biological labelling, solar cells, transistor components and electrochemical sensors [49,50]. Organic counterparts, such as the carbon quantum dot, graphene quantum dot, and the graphene oxide quantum dot, have received tremendous attention in the design of electrochemical sensors because of their relatively small size, large surface area, lower toxicity and readily available precursors [51,52].

Two broad pathways can synthesize carbon-based quantum dots, such as carbon and graphene quantum dots,: top-down and bottom-up routes. Top-down approaches, such as chemical ablation, plasm treatment, laser ablation, electrochemical synthesis and thermal decomposition, involve the breaking down of larger materials such as graphene and carbon nanotubes into smaller components [53]. Conversely, bottom-up routes entail the formation of quantum dots from small molecules such as organic acids or big molecules like carbohydrate using chemical, plasma, hydrothermal and solvothermal treatment [53]. More interesting is the fact that these carbon-based quantum dots can be produced from cheap readily available materials such as starch [54], pomelo peel [55], water melon peel [56], citric acid [57], hair fibre [58], grass and leaves [59,60]. Beyond the availability of these cheap precursors, carbon-based QDs have a large surface area, high chemical stability, good conductivity and excellent optoelectronic properties [61,62]. These advantages have made the deployment of these class of materials in the design of electrochemical sensors [51] and fluorescence probes [62,63] alluring, as is evident from the numerous studies available. Carbon-based quantum dots have been used in the design of electrochemical sensors for monoamine neurotransmitters with reliable sensitivity and selectivity in the presence of other interfering molecules [51,64].

Advancements in the determination of various neurotransmitters using nanomaterials for optical and electrochemical sensor design and the prospect of these techniques have been reviewed by Chauhan et al. [65]. In the same vein, the electrochemical determination of NTs and the mechanism of electron transfer was discussed in a review put together by Arumugasamy et al. [66]. A review specific to the challenges involved in the detection of DA in biological samples using chemically modified electrode was done by Sajid et al. [67] with special reference to numerous carbon-based electrodes, ionic liquids, polymers, and metal and metal oxide nanoparticles. However, Sajid and his group of researchers showed little reverence for the importance of quantum dots. Ribeiro et al. [68] put good effort into discussing the progress made in the design of electrochemical sensor and biosensors for the detection of neurotransmitters over fifteen years (2000–2015), with special attention on the various materials applied for the production of the sensors. Ribeiro and his crew did not refer to the application of quantum dots in their lengthy review. Tajik and his group [69] presented a review on the synthesis, application of CQDs and GQDs in electrochemical sensing of a wide range of biomolecules, as well as the characterization of QDs. Li et al. [70] reviewed the synthesis and the application of CQDs and GQDs in electrochemical, chemiluminescence, photoluminescence and optical sensors over the last five years. Advantages and the disadvantages of using CQDs and GQDs and their application as surface modifiers for biological molecules (not neurotransmitters) from 2015–2019 were presented by Campuzano et al. [71]. Accordingly, the efforts that have gone into the determination of monoamine neurotransmitters using carbon-based quantum dots (CQDs and GQDs) were discussed in this review with special focus on the source of the QDs precursor, limit of detection, linear dynamic range and selectivity in the presence of interfering biomolecules. We further delved into the comparison between these two QDs and the performances of each type of carbon-based QD relative to another, especially in DA detection. A general scheme for the redox reaction of catecholamine and the indolamine NTs was also proposed in this study. Graphene QDs and carbon QDs were the most common of the lot; as such, only their performance in the detection of monoamine neurotransmitters has been compared in the last ten years. To the best of our knowledge, this is the first time a review of this nature has been attempted.

## 2. Graphene and Carbon Quantum Dots: Similarities and Disparities

### 2.1. Similarities between CQDs and GQDs

Graphene and carbon QDs belong to the class of zero-dimensional nanomaterials—quantum dots [69]. They both possess sp^2^ hybridized carbon atoms arranged to form a graphitic core [72,73]. Just like their inorganic counterparts, they possess quantum confinement, which makes them different from ordinary graphene and also contributes to their optical properties [74]. For both carbon and graphene QDs, their optical properties vary with size and shape. They possess low cytotoxicity which places them above the inorganic semiconductor class of quantum dots [71] in electrochemical sensors preparation. The biocompatibility of both CQDs and GQDs have been put to test with satisfactory results. For instance, Lim et al. [75] carried out one of the early electrochemical detection of biomolecules with GQDs and CQDs and found out that they can favourably be applied for the detection of ascorbic acid (AA). This can be ascribed to the presence of carboxylic, hydroxyl or amino group on their edges which makes them soluble in aqueous medium [71]. They have both been used as support for other materials and as signal transducers in electrochemical sensors because these surface functional groups (which vary based on formation route) enhance their modification with inorganic materials, organic species and enzymes. This is evident in the modification of CQDs with ionic liquid modified graphene (IL-graphene), as presented by Zhuang et al. [76] in 2016. In that study, the ionic liquid conferred a positive charge on the graphene sheets, which led to the electrostatic attraction for the CQDs with negatively charged edge functionalities. In the same vein, 3-aminophenyl-boronic acid has been applied for the chemical modification of N-CQDs [62]. Similar occurrence prevailed with GQDs when modified with room temperature ionic liquid (RT-IL) for CPE modification [77].

Furthermore, these classes of quantum dots (GQDs and CQDs) have good optoelectronic properties, which makes them applicable in numerous processes where fluorescence is required. This has been demonstrated in various fluorescence sensors for the detection of biomolecules. Research shows that they have better photostability against photobleaching than some of the inorganic semiconducting QDs [70]. It is equally noteworthy that these QDs cost much less to prepare compared to the metallic semiconductor QDs due to the readily available cheap precursors. Additionally, doping these two types of QDs with heteroatoms improves their electrocatalytic activity towards the oxidation of biomolecules. Specifically, N-doping has been confirmed to improve the electrical conductivity and chemical stability of GQDs [62,78]. This development was further confirmed by Jiang et al. [62] while discussing the merit of doping carbon nanomaterials with a heteroatom such as nitrogen. Jiang and his group emphasized that the fact that N-doping will confer induced charge delocalization on such nanomaterials, suggesting same fate for CQDs. Going by the surface morphology of CQDs and GQDs, certain similarities can also be established. Specifically, judging by the internal structure of these QDs from the transmission electron micrograph (TEM) of CQDs and GQDs produced from different precursors as seen on Figure 3, it can be inferred that both GQDs and CQDs have particles of either spherical or round shape under the electron microscope with particle sizes spread over few diameters, regardless of the source of the carbon-based quantum dots. Even when doped with a heteroatom such as nitrogen, the round or spherical shape is still maintained (Figure 3E,F).

### 2.2. Differences between GQDs and CQDs

Despite the numerous similarities between these QDs, differences still abound. CQDs are amorphous materials often produced from biomass in a bottom-up approach. GQDs are mostly crystalline and often made from graphene or graphite in a top-down approach [73], such that the resultant QDs still retain some features of graphene. In terms of size, GQDs and CQDs have been reported to have sizes of less than 100 nm and 10 nm, respectively [75,85], but the particle sizes obtained from some recently synthesized GQDs have been evaluated to be close to those of some CQDs. This in a way makes it difficult to differentiate these QDs with particle sizes only. For instance, GQDs of ~3 nm average diameter was synthesized by Tashkhourian et al. [86], while a bigger particle size of ~5 nm was reported by Zhuang et al. [76] for CQDs. GQDs are less than ten layers of graphene [69] but the CQDs are simply carbon nanomaterials with graphitic core [73].

The similarities and differences between these two classes of QDs can be found on Table 1 at a glance.

## 3. Electrochemical Methods

The accurate detection of an analyte in an electrochemical system depends on the electrochemical method adopted. This is because the various electrochemical techniques have different levels of sensitivity. However, some techniques have other ways of making up for their relatively low sensitivity. For instance, cyclic voltammetry, though with relatively low sensitivity can satisfactorily be applied for the determination of the mechanism and kinetics of a redox reaction at the surface of a working electrode. In this section, of the review, the electrochemical methods used for the detection of the monoamine neurotransmitters will be thoroughly discussed.

### 3.1. Cyclic Voltammetry

This is about the most applied electrochemical technique. It gives information about the mechanism and kinetics of the reaction going on at the surface of the working electrode. It entails imposing a potential difference over a specific range known to contain the formal potential of the analyte [87]. The current response of the analyte, which often has a linear relationship with the concentration of the analyte, is used for analytical purposes. The potential is swept to the positive side from a starting potential (*E_s_*) to the final potential (*E_f_*) for an oxidation reaction followed by an immediate sweep in the opposite direction from the final to the original starting potential (Figure 4A) [88]. This potential sweep occurs over a specific time frame. The fraction of the change in potential (ΔE) to the change in time (Δt) gives the scan rate (*v*). The scan rate has an impact on the current response of the analyte [89]. In most cases higher scan rate is preferable. The oxidation peak potential (*E_pa_*) is the potential where the highest oxidation current (anodic peak current, *i_pa_*) is obtained, while the converse is true for the cathodic peak current (*i_pc_*) (Figure 4B). The difference between the peak potentials gives an idea about the reversibility of the reaction. Specifically, the closer this value is to a threshold of 59/*n* mV (at 25 °C) and the closer the ratio *i_pa_*/*i_pc_* is to unity, the better the reversibility [90].

In principle, the transport of the electroactive species from the bulk solution to the electrode–electrolyte interface (mass transport) could be controlled by migration, diffusion or convection [87]. Migration is a product of the chemical or physical attraction of the analyte for the electrode and this often leads to surface adsorption of the analyte to the electrode. This possibility affects the reversibility of the reaction adversely. Convection, which comes into play by stirring of the electrolyte and analyte mixture, is often eliminated. Diffusion is the most desirable because it makes available the oxidized and the reduced form of the analyte at approximately the same concentration. This often gives a reversible or near-reversible reaction [88].

The Randle-Sevcik equation, Equation (1), suggests that a linear relationship exists between the peak current (*i_p_*) of a freely diffusing reversible electrochemical redox specie and the square root of its scan rate (*√v*) [90]. This is usually investigated by plotting *i_p_* against *√v*. Deviation from this relationship could mean quasi-reversibility or outright irreversibility (surface adsorption). Through this equation, the diffusion coefficient, as well as the electroactive specie surface coverage (Γ (mol cm^−2^)), can be obtained. The other form of the Randle-Sevcik equation, Equation (2), is often used to obtain the surface coverage by plotting *i_p_* against *v* [87]. In the equations, *i_p_* (A), *n*, *F* (C mol^−1^), *v* (Vs^−1^) and *A* (cm^2^) represent peak current, the number of electrons transferred, Faraday’s constant and electrode surface area, respectively. *D*_0_ (cm^2^ s^−1^) is the diffusion coefficient of the oxidized analyte while the °C (mol cm^−3^) is the bulk concentration of the analyte. *R* (J.mol^−1^ K^−1^) and *T* (K) represent the usual molar gas constant and absolute temperature, respectively.
(1)$$i_{p}=0.446nFAC^{0}{\left(\frac{nFvD_{0}}{RT}\right)}^{1/2}$$
(2)$$i_{p}=\left(\frac{n^{2}F^{2}}{4RT}\right)vA\Gamma$$

Notably, a variant of Equations (2) and (3) can be easily used to calculate the number of electrons transferred, provided that the quantity of charge integrated from the area of the peak, *Q* is known. This can be achieved from a plot of *i_p_* against *v* from which the slope can be used to obtain the value of electrons transferred (*n*).
(3)$$i_{p}=\frac{nFQv}{4RT}$$

The charge transfer rate constant (*k_s_*) can be obtained by the dependence of *E_p_* on the scan rate, as described by Laviron [91]. The plot of either the *E_pa_* or the *E_pc_* against ln*v* gives a slope from which the charge transfer coefficient, α is calculated, provided the number of electrons transferred (*n*) can be obtained. This works for a quasi-reversible surface-confined processes that are also suspected to be partly diffusion controlled.
(4)$$E_{pa}=a+\left(\frac{RT}{\left(1-\alpha \right)nF}\right)\ln v$$
(5)$$E_{pc}=b-\left(\frac{RT}{\alpha nF}\right)\ln v$$
(6)$$logk=\alpha \log\left(1-\alpha \right)-\log\left(\frac{RT}{nFv}\right)-\;\alpha \left(1-\alpha \right)\frac{nF\Delta E}{RT}$$

For a surface-confined irreversible process at 25 °C, Equation (4) can be substituted into Equation (3) to obtain Equation (6), from which *n* can be obtained. The value of αn is obtained directly from Equation (5). Therefore, the electron transfer coefficient α can be obtained.

The equations above have been used for establishing the electrochemical behaviour of the quantum dots-based electrodes used in the determination of the monoamines as well as the kinetics of the redox reactions at the surface of these electrodes. Specifically, the electrochemical behaviour of the quantum dots modified electrodes were investigated with the use of [Fe(CN)_6_]^3−/4−^ redox probe in CV experiments [80,86,92,93]. The anodic and cathodic peak current ratios were sometimes used to estimate the reversibility of the redox process while the electroactive surface area was calculated from the known value of *D_o_*, *C^o^* and *n* using Equation (1). Huang et al. [80] used this equation to obtain the value of the CQD-modified electrode surface area in the presence of K_3_Fe(CN)_6_ in 0.1 M KCl using CV. Just as reported for most modified electrodes, the CQD-modified electrode had a higher surface area than the bare electrode. A similar trend was observed using CQD-modified GCE fabricated by Li et al. [94] for DA detection and GQDs/2D-hBN/GCE prepared by Yola and Atar [95] for SE detection.

In most literature, the reversibility of the reactions was estimated from the anodic and cathodic peak potential difference obtained in the presence of the monoamine neurotransmitter or the current ratio. For instance, the reactions of DA at the surface of GQDs-NHCH2CH2NH)GCE was determined to be quasi-reversible by Li et al. in 2016 [96], largely because of a potential peak difference of 0.087 V obtained. A Similar trend was observed in some other literature where QDs have been used for the modification of the signal transducer [84]. Ruiyi et al. [97], Zheng et al. [81] and Li et al. [96] reported almost reversible processes at their respective QDs-based electrodes in CV experiments with DA where peak potential differences of 0.03, 0.025 and 0.031 V, respectively, were obtained. EP redox reaction on CQDs/CPE prepared by Shankar et al. [93] was also reported to be quasi-reversible due to the large anodic and cathodic peak difference of 0.132 V obtained. The modified electrodes are often adjudged better than the bare electrodes, since a much lower potential peak difference is often obtained for the QD-modified electrodes. In some cases, the CV peak current of the QD-modified electrode compared to that of the bare electrode in the presence of the analytes or [Fe(CN)_6_]^3−^/^4−^ redox probe in PBS solution was used to establish the better electrochemical characteristics of the carbon-based QD-modified electrode [52,76,79,97,98,99].

The variation of the square root of scan rate with the peak currents was applied for obtaining the mechanism of the redox reactions at the surface of the quantum dots modified electrodes. A direct proportion implies a diffusion-controlled reaction [90]. The peak current of EP on the CQD-modified GCE fabricated by Shankar et al. had a linear relationship with the square root of the scan rate. Consequently, the process was reported as diffusion controlled. This was also evident in the behaviour of DA at the Au@CDs-CS-modified electrode fabricated by Huang et al. [99] in 2014, where the square root of the scan rate increased with the peak currents. A similar trend was observed by Tashkhourian et al. [86] when they applied GQD-modified GCE for EP detection. On the other hand, the direct proportion between the peak current (or logarithm of peak current) and the scan rate (or logarithm of scan rate) suggests an adsorption-controlled reaction. A good example of this was found at the Au-GQDs-Nafion modified GCE prepared by Jang et al. in 2019 [98] as well as CD-CQDs/GCE presented by Chen et al. in 2017 [100]. Another surface-controlled process was reported by Jiang et al. [83] as a result of an increase in scan rate with increasing peak currents obtained at NCQDs/GCE just like adsorption-controlled process reported by Huang et al. [82] for Cu_2_O-CDs/nafion composite-modified GCE. CQD-modified SPCE prepared by Devi et al. [101] for DA detection followed this same trend. Li et al. [94] equally suggested a surface adsorbed phenomenon at the CQD-modified GCE when applied for DA detection going by the increase in log *v* with an increase in log *i_p_*. In some instances, both the surface adsorption and the diffusion phenomenon can be operational in a particular redox reaction. In such cases, the linear relationship between *i_p_* and v at lower scan rate persists between *i_p_* and the square root of *v* at a higher scan rate. This was the case with EP at the surface of the GQDs-CS modified CPE fabricated by Tashkhourian et al. in 2018 [86].

Few of the carbon-based QD-modified electrodes were used for investigating the electron transfer coefficient (*a*) and the heterogeneous rate constant (K_s_) of the redox process of the monoamine NTs using Equations (4)–(6). One of the electrodes applied in this regard is the Au@CDs-CS modified GCE presented by Huang et al. in 2014 [99]. Huang and his co-workers estimated *a* and *K_s_* as 0.40 and 0.22 s^−1^, respectively. Cu_2_O-CDs/nafion/GCE fabricated by Huang et al. in 2015 [82] was also used for estimating these values as well with Ks and *a* values of 1.48 s^−1^ and 0.603, respectively obtained. Alternatively, the relationship between the peak potential and the logarithm of scan rate, Equation (4), was used to obtain the value of *a*. Shankar et al. [93] used this approach to obtain the value of *a* as 0.43 for CQDs/GCE from the slope of a plot of *E_p_* against log *v*.

The sensitivity of CV is hampered by the significance of charging current. However, its importance in the determination of kinetics and mechanisms of electrode reactions makes it one of the most important electroanalytical techniques. The analytical application of the electrodes was mostly carried out with the aid of more sensitive techniques such as DPV and SWV.

### 3.2. Differential Pulse Voltammetry (DPV)

Apart from the current flow between the electrode and the analyte in solution as a result of electron transfer (faradaic current), there is also a possibility of current flow without such transfer of electron [102]. This occurs as a result of the formation of a double layer capacitance between the electrolyte solution and the electrode. This relationship generates a charging current (non-faradaic current) that limits the faradaic current recorded in cyclic voltammetry because the total current output is a summation of the duo [103]. Pulse voltammetry has greatly reduced this charging current by progressively applying short pulses over time. These pulses are assigned within a potential step such that the current before the end of a pulse (*I*_1_) and just before the application of a pulse (*I*_2_) is subtracted to obtain the current output in a pulse (Figure 5A) [104]. The cumulative current over a certain potential range is plotted against the potential to obtain a differential pulse voltammogram (Figure 5B) [104]. The magnitude of the charging current, in this case, is so small that the DPV gives a better result in terms of sensitivity compared to CV. This improved sensitivity is the reason DPV is mostly used for selectivity test in electrochemical sensors. The outstanding sensitivity makes discrimination of analytes feasible.

As a result of this proven sensitivity of DPV, more than 90% of the carbon-based QD-modified electrochemical sensors have been used for analytical determination of monoamine neurotransmitters via DPV. Additionally, the real sample determination and selectivity tests have been mostly carried out with the help of DPV which makes signal detection in the presence of interferents much easier. Only a few publications have attempted the use of other techniques for the analytical application of the fabricated electrodes.

### 3.3. Square Wave Voltammetry

Square wave voltammetry (SWV) combines the good qualities of sweep voltammetry, such as CV, and pulse voltammetry. This is so because the SWV can be used for some mechanistic deductions of the electrochemical process while giving a better sensitivity (than CV) at a faster scan rate than the DPV [102]. SWV is faster than DPV and more sensitive than the CV. Similar to the DPV, potential pulses within a step are also imposed while the step increases progressively with time. The current responses are recorded at the end of the pulse in the forward direction (*I*_1_) at the forward potential (for oxidation) and at the end of the pulse in the backward direction (for reduction) (*I*_2_) (Figure 6A). The current response is plotted against the potential in each case (oxidation and reduction) to obtain a replica of the cyclic voltammogram (Figure 6B) [104,105]. These cumulative current responses for each direction are subtracted to obtain the net current which is often greater than the individual currents. This is because the charging current for the forward and the backward reactions are similar in magnitude and stand a chance of cancelling out after subtraction [104].

Before subtraction, the forward current–potential curve and the reverse current–potential curve have the look of a CV which can be used for investigating the mechanism of the reaction. The SWV, as a result of its sensitivity, has been used for the determination of analytes where high sensitivity is required. It has equally been used for the selective determination of analytes in the presence of interferents as well as the simultaneous determination of analytes.

The SWV did not enjoy enough patronage in the detection of monoamine neurotransmitters using carbon-based QD-modified electrodes as the DPV. Specifically, the only publication where this technique was used for analytical detection was in NE determination with GQDs-AuNPs/GCE [106]. The technique used was not a direct SWV but the square wave stripping voltammetry (SWSV). The SWSV is a slight modification of the SWV with an accumulation phase where the analytes are accumulated on the electrode for a few seconds before being stripped. This configuration improves the availability of the analyte on the electrode and is, therefore expected to give a better current response.

### 3.4. Linear Sweep Voltammetry

Square wave voltammetry is a relatively fast technique amenable to very fast reactions. However, it offers a small response to sluggish chemical reactions compared to a simpler method such as the linear sweep voltammetry (LSV) [102]. LSV involves scanning over a fixed potential from a lower to an upper limit over some time in order to obtain a voltammogram of the current response against voltage (Figure 7A,B) [107]. The scan is done from the lower limit through a voltage where the peak current is obtained to the upper limit where the current eventually falls [108]. Just like the CV, the current response changes with change in the scan rate for fast reversible electron transfer processes at a similar potential. Slow irreversible processes show a decline in current response and shift in the position of the electrode potential.

The detection of monoamine NTs with carbon-based QD-modified electrodes via LSV was attempted only once. This was only seen in DA detection with CQD-modified GCE fabricated by Algarra et al. in 2018 [64]. The electrode was deployed for the analytical determination of DA without establishing the LOD of the modified electrode.

### 3.5. Chronoamperometry (CAP)

Chronoamperometry is another potential step technique where the potential step is applied between initial potential *E*_1_ (when no current is flowing) and the final potential *E*_2_ while the current response is measured with time. The current flow at any time after the application of *E*_2_ is expected to obey the Cottrell equation, Equation (7), for a diffusion-limited process. The voltammogram obtained from chronoamperometry is a plot of the current response against time (Figure 8). CAP can be used for studying the kinetics and mechanism of chemical processes [110].
(7)$$i\left(t\right)=\frac{nFAD_{o}{}^{1/2}C^{o}}{{\left(\pi t\right)}^{1/2}}$$

All variables in Equation (7) retain the definitions earlier given.

Chronoamperometry was used for the analytical determination of EP using the GQD-modified CPE prepared by Tashkhourian et al. [86] with very low LOD (0.3 nM) reported for the electrode. This was the only time CAP was applied for analytical determination of monoamines with a carbon-based QD-modified electrode. With the aid of Cottrell’s equation Equation (7), Jahani [111] calculated the diffusion coefficient, D_o_ as 1.6 × 10^−6^ cm^2^/s from the experimental value of the current, *i* and square root of time, *t* obtained with the application of the fabricated GQDs/IL/CPE for NE detection.

## 4. Electrode Fabrication

### 4.1. Modification of Glassy Carbon Electrodes (GCE)

Glassy carbon electrodes (GCEs) are one of the most adopted working electrodes for electrochemical sensors fabrication due to its favourable chemical properties such as chemical inertness, little liquid permeability and small pore size [112]. These qualities might have been the reason the greatest percentage of electrodes modified with carbon-based quantum dots for monoamine NTs detection were GCE. Only a few such electrodes are screen printed electrodes and carbon paste electrodes. In this section of the review, the fabrication of the various modified glassy carbon electrodes will be discussed.

Pang and co-workers [79] modified a GCE with GQDs prepared from graphene oxide (GO) using the hydrothermal technique. Notably, the GO precursor was synthesized from graphite powder via the hummers’ method. The GQD-modified GCE was fabricated by simply casting the GQDs suspension on the hitherto cleaned GCE and dried for 30 min at about 50 °C.

Li et al. in 2016 [96] fabricated another GQD-modified GCE with a slightly different architecture. The GQDs were prepared from the carbonization of a citric acid precursor and dropped on the surface of a clean GCE functionalized with ethylenediamine. The functionalization of the GCE was done by initial carboxylation and immersion in ethylenediamine. The resultant electrode was labelled GQDs-NHCH_2_CH_2_NH)/GCE.

In 2019, Hsu and Wu [84] modified GCE with N-doped GQDs, SnO_2_ nanoparticles and polyaniline to obtain a working electrode tagged SnO_2_/PANI/N-GQD/GCE. The N-GQDs were prepared by heating a mixture of citric acid and urea while the SnO_2_ nanoparticles were prepared through the hydrothermal process. The SnO_2_/N-GQDs/PANI composite was prepared from the oxidative polymerization of aniline in the presence of SnO_2_ nanoparticles, relying on the electrostatic attraction between the negatively charged N-GQDs and SnO_2_/PANI composite when the dual composite was mixed with N-GQDs. The SnO_2_/PANI/N-GQD composite was washed with deionized water, dried at 60 °C and cast on GCE to obtain the modified working electrode.

Gold nanocrystals (AuNCs) prepared through an environmentally friendly technique in the presence of TMSPED were capped with GQDs prepared from glucose for GCE modification in 2019 by Vinoth and his team [52]. The TMSPED was introduced into the gold nanocrystals to prevent the aggregation of the AuNCs and to provide the functionalities necessary for the covalent linkage of the AuNCs to the GQDs. In essence, the TMSPED-AuNCs composite prepared via ultrasonic means were irradiated with GQDs to obtain the GQDs-TMSPED-AuNCs nanocomposite cast on GCE.

Histidine (His)-functionalized GQDs (His-GQDs) and graphene micro-aerogel (GMA) hybrid were used for the modification of GCE by Ruiyi and his colleagues [97] in 2017 (Figure 9). His-GQDs were prepared through pyrolysis of histidine and citric acid. This was followed by the addition of Zn^2+^ to His-GQDs to form a complex which acted as a surfactant for the GO dispersion which was further reduced by hydrazine hydrate to form the graphene micro-gel. It is important to note that the ultrasonic dispersion of Zn-His-GQDs in the GO was made easier by the introduction of toluene into GO to form an emulsion largely stabilized by the Zn-His-GQDs. The micro-gel was subjected to acid washing, freezing and thermal annealing to form the His-GQDs-GMA which was subsequently dispersed in ethanol to form a dispersion that was dropped on the surface of a clean GCE. The modified GCE was dried in air before use.

Arumugasamy et al. [113] modified a GCE with GQDs and acid-functionalized MWCNTs for DA detection in 2020. The MWCNTs sonicated in HCl was dispersed in deionized water and added to GQDs made from glucose to form a homogeneous GQDs-MWCNTs suspension subsequently drop-casted on clean and pre-treated GCE. The resultant electrode was tagged GQDs/MWCNTs/GCE. It was expected that the electrode would benefit from the conductivity of the GQDs and the MWCNTs.

Intending to present a very sensitive electrode with good electrocatalytic activity, Jang and co-workers [98] combined AuNPs and GQDs made from the hydrothermal treatment of carbon black to form a composite that was later cast on GCE (Figure 10). This was followed by drying the modified GCE under an infrared lamp before coating with a portion of diluted nafion. The nafion coated working electrode (Au-GQDs-Nafion/GCE) was further dried under an IR lamp before application for the electrochemical detection of DA.

In 2018, Zheng et al. [81] fabricated a GQD-modified GCE by the electrodeposition of GQDs made from the pyrolysis of citric acid on a previously cleaned GCE. The electrodeposition was done with CV over a potential window of −1.5–2.0 V at a scan rate of 0.1 Vs^−1^ for about 30 cycles. The resultant electrode was designated GQDs/GCE. Specifically, this was the only electrochemically deposited GQD-modified GCE used for DA detection over the years under review.

GQDs were also used to modify GCE to obtain sensors capable of detecting other monoamine neurotransmitters. Baluta and his team [114] developed one of such electrodes by modifying GCE with GQDs obtained from the pyrolysis of citric acid at 200 °C. Prior to coating the surface of the thoroughly cleaned GCE with the GQDs, the GQDs were modified with enzyme laccase. To enhance the physical adsorption of the enzyme onto the surface of the electrode, laccase was crosslinked with glutaraldehyde (GA). The modified electrode tagged GCE/GQDs/Lac was washed with buffers to remove unbound enzyme moieties. The crosslinked laccase on the surface of the GQDs activates the electrode and could remain active for another 4 months.

GQDs made from graphite and functionalized with 2-aminoethanethiol (AET) was ultrasonicated along with 2D-hexagonal boron nitride (2D-hBN) by Yola and Atar [95] for GCE modification. 2D-hBN was prepared by heating a mixture of boron nitride powder and isopropyl alcohol for 24 h at 50 °C, followed by ultrasonication. The precipitate obtained after centrifugation was washed and dried at about 60 °C to obtain the 2D-hBN. The GQDs/2D-hBN composite was dropped on the surface of a clean GCE to obtain the modified GCE working electrode for SE detection.

Fajardo et al. [106] synthesized another GQDs from the pyrolysis of citric acid (Figure 11) for GCE modification alongside AuNPs. Gold nanoparticles (AuNPs) were prepared by heating a mixture of cyclodextrin, NaOH and HAuCl_4_ at 120 °C for about 4 h. The GCE to be modified was cleaned and activated electrochemically with CV scan over a potential range of 0–2 V at a scan rate of 50 mVs^−1^ for ten cycles. This was followed by immersing the activated electrode in poly(diallyldimethylammonium chloride), rinsed with water, immersed in GQDs solution and finally coated with AuNPs to obtain GCE/GQDs/AuNPs which was used NE detection.

Just like the GQDs were used for GCE modification, several CQD-modified GCE were also fabricated for monoamine NTs detection. An instance was given by Chen et al. [100] in their laboratory where poly(β-cyclodextrin) and CQDs were used for GCE modification. A mixture of CQDs prepared from the pyrolysis of citric acid and the readily available β-cyclodextrin mixture was subjected to CV scan over a potential of −1.0–1.0 V in the presence of the cleaned GCE to form the poly(β-cyclodextrin)/CQDs composites deposit on GCE. The modified electrode was subsequently washed distilled water and dried in air at room temperature. For comparison, other variants of the electrode such as β-cyclodextrin/GCE and β-cyclodextrin/CQDs/GCE were fabricated.

Canevari et al. [51] prepared a GCE modified with GO, SWCNTs and an electrochemically synthesized CQDs for DA detection. The CQDs were prepared from 1-propanol and KOH solution subjected to a potential and current of 6.5 V and 100 mV, respectively. These CQDs, together with GO and SWCNTs were applied individually for coating the surface of a pretreated GCE. The resultant electrode was left to dry overnight before being applied for the determination of DA.

Microwave-assisted CQDs preparation from glucose and polyethylene glycol provided the CQDs added to chitosan (CS) by Huang et al. in 2013 [80] for GCE modification. The CQDs were added to CS with ultrasonication before casting the composite on GCE. The modified electrode was dried at 60 °C for about 30 min to obtain the CQDs-CS/GCE for DA detection.

Zhuang and his colleagues in 2016 [76] designed GCE modified with ionic liquid functionalized graphene (IL-graphene) and CQDs. The IL-graphene and the CQDs were mixed to form a solution which was autoclaved for 4 h at 90 °C before being dropped on the surface of the polished GCE. The resultant modified GCE was dried at room temperature and rinsed with distilled water to remove adsorbed impurities on the surface of the electrode.

Using N-doped CQDs (N-CQDs) prepared from ammonia and collagen via hydrothermal means for GCE modification, Jiang et al. [83] presented another carbon-based QD-modified GCE for DA detection in 2015. The N-CQDs were initially modified with nafion solution to obtain N-CQDs-nafion solution which adhered better to GCE compared to the unmodified N-CQDs. The modified electrode was dried under ambient condition before use. Nafion was introduced to serve as a binder of the N-CQDs to the polished GCE surface and also to aid the discrimination of interferents such as AA and UA so that DA can be selectively detected.

Another N-CQD-modified GCE was prepared by Jiang et al. in 2015 [115] as well. The major difference between this electrode and the previous one is that the N-CQDs were prepared through a faster microwave-assisted means using a single diethanolamine (DEA) precursor. After synthesis in the microwave reactor, the N-CQDs were frozen at −80 °C and dried using vacuum before eventual casting on the initially cleaned GCE.

CQDs made from the carbonization of glucose after ultrasonication of the glucose solution were used for GCE modification by Li et al. in 2015 [94]. The GCE was mechanically polished, washed with sulphuric acid and activated electrochemically with CV over a potential range of −0.2–1.5 V before the drop-casting of CQDs. The modified GCE was dried in air overnight before application for DA detection.

By heating the mixture of sucrose and oil acid at 215 °C and subsequent extraction of the oil acid from the sucrose and oil acid precipitate dissolved in water, CQDs for subsequent GCE modification alongside Au nanoparticles and chitosan were prepared by Huang and co-workers in 2014 [99]. The CQDs were added to HAuCl_4_ to form a solution that was heated at 100 °C to obtain the AuNPs-CQDs composite. The solution of this composite was added to chitosan (CS) with constant ultrasonication to form a solution that was eventually cast on GCE. The modified electrode was dried in the oven for 30 min at 60 °C.

Again, Huang et al. [82] proceeded to apply the same CQDs earlier manufactured from the carbonization of sucrose and oil acid for GCE modification in 2015. This time, Cu_2_O nanoparticles and nafion were used in place of AuNPs and chitosan, respectively. Huang and co-workers added the CQDs to a solution of Cu(NO_3_)_2_ and NaOH and heated the mixture at 90 °C to obtain the Cu_2_O-CQDs composite dispersed in a solution of ethanol and nafion with constant ultrasonication. The Cu_2_O-CQDs-nafion nanocomposite was coated on a clean GCE surface to obtain the modified working electrode.

The only carbon quantum dot prepared through electrochemical exfoliation of graphite for GCE modification was made available by Devi et al. in 2018 [101]. Two graphite electrodes were subjected to a constant direct current of 50 mA in a solution containing NaOH, ethanol and water. The electrochemical cutting of the graphite was induced by the migration of hydroxyl ion to the anode where oxidation occurred. The CQDs produced were washed and stored at 4 °C prior to coating on the surface of clean GCE and screen-printed carbon electrode. Nafion was further coated on the surface of the GCE/CQDs and SPCE/CDs to enhance their selectivity for DA in the presence of AA and UA. The two electrodes were used for the analytical determination of DA.

Molecular imprinting technique was adopted by Yola and Atar in 2019 [116] to prepare an EP imprinted polypyrrole on GCE initially modified with graphitic carbon nitride (g-C_3_N_4_) and N-CQDs. g-C_3_N_4_ prepared from the thermal condensation of melamine while N-CQDs were made from the carbonization of a mixture of citric acid and urea. A composite of these two (g-C_3_N_4_/N-CQDs) was made by heating a mixture of the N-CQDs and melamine at 600 °C for about 4 h. This was followed by dropping a dispersion of the composite on the surface of GCE after which the solvent was removed with IR lamp. This modified GCE was used in CV experiments with PBS solution containing EP and pyrrole for 25 cycles with the aim of electropolymerizing pyrrole with EP imprinted. The resultant electrode was tagged MIP/g-C_3_N_4_/N-CQDs/GCE.

### 4.2. Modification of Carbon Paste Electrode

Carbon paste electrodes (CPE) are popular for their cost-effectiveness, low background current and low ohmic resistance [117]. These properties and the ease with which electrode modifiers can be added to the graphite powder and the pasting liquid during the fabrication process of the electrode made its application as electrochemical sensors more appealing. After GCE, whcih was extensively modified with carbon-based quantum dots for monoamine NTs detection, the CPE came next. This section discusses the fabrication of carbon-based quantum dot-modified CPE for monoamine NTs detection. Unfortunately, carbon-based modified CPE was not used for DA detection over the years under review.

Tashkhourian et al. [86] fabricated GQD-modified CPE for EP detection in 2018. Through the simple and common citric acid carbonization, GQDs were synthesized and mixed with graphite powder-paraffin oil adduct (CPE) and chitosan solution prior to ultrasonication for 15 min and drying at 80 °C. Silver wire was passed into the plastic tube housing the GQDs, graphite and chitosan to establish electrical contact. The surface of the further smoothened with fax paper in readiness for analytical application.

Jahani [111] prepared GQDs and ionic liquid composite for CPE modification in 2020 for NE detection. This was accomplished by mixing the GQDs, graphite powder and the IL in a mortar and pestle. The resultant paste was packed into a glass tube through which a copper wire was passed to establish electrical contact. To establish the significance of the GQDs and the ionic liquid, IL-CPE and GQDs-CPE were fabricated and characterized electrochemically along with the actual working electrode (GQDs/IL/CPE). Similar electrode (RTIL-GQDs/CPE) was prepared by Sanati et al. [77] for levodopa detection in the presence of SE. The IL, in this case, was a room-temperature ionic liquid (RT-IL) containing 1-butyl-3-methylimidazolium hexafluorophosphate. This IL was mixed with graphite powder, GQDs made from the pyrolysis of citric acid and paraffin. The resultant paste was transferred to a glass tube through with copper wire was passed for electrical contact with the potentiostat.

CPE was also modified with CQDs by Shankar et al. in 2019 [93]. The CQDs were prepared by burning styrene in air followed by stirring for 25 min in NaOH solution. This solution was ultrasonicated and subsequently filtered to obtain CQDs. The CQDs was then mixed with graphite used for the preparation of the CPE.

### 4.3. Modification of Screen-Printed Electrodes

Screen-printed carbon electrodes are economical signal transducers with small size, which makes them good components for miniaturized electrochemical sensors. They are composed of a working, reference and counter electrode in a compact system. The conductive carbon ink doubles as the working and counter electrode while silver tracts serve the purpose of a reference electrode [118]. SPCE can be easily produced in large quantity. This contributes to their wide application in the fabrication of sensors for analytes in various matrices.

SPCE modified with CQDs prepared by Devi et al. was discussed in the previous section. Two other carbon-based quantum dot-modified screen-printed electrodes were fabricated for monoamine NTs detection within the years under review. Specifically, those electrodes were targeted towards DA detection alone. One of such electrode was made available by Ben Aoun in 2017 [92]. He prepared N-GQDs from the microwave irradiation of glucose and ammonia at 300 W for about 5 min. These N-GQDs were mixed with chitosan solution and vigorously stirred to obtain the Chitosan/N-GQDs used for SPCE modification. Prior to modification with this composite, SPCE was electrochemically activated by CV scan between −1–1 V at a scan rate of 100 mVs^−1^. The modified electrode was designated CS/N-GQDs/SPCE. The other carbon-based QD-modified SPCE was fabricated by Beitollahi et al. in 2018 [119]. Therein, GQDs prepared through the pyrolysis of citric acid in a ceramic crucible were dispersed in water, ultrasonicated for 30 min and subsequently cast on SPCE. The solvent was evaporated to give the GQDs/SPCE that was eventually applied for DA detection.

## 5. Mechanism of Monoamine Neurotransmitters Detection

Regardless of the nature of the electrode, the detection of Catecholamine neurotransmitters and indolamines follows the simple redox pathway. The catecholamine NTs undergo oxidation by losing the active hydroxyl hydrogen of the catechol hydroxyl group to obtain the quinone form of the analyte while the reduction takes place by the replacement of the hydrogen groups to revert to the catechol form (Scheme 1). While these happen across the surface of the modified electrode, it is essential to understand that the modified electrode serves as an electron pump capable of causing the reduction of the oxidized form of the analyte, while electron loss from the reduced form occurs due to the potential difference between the reduced form of the analyte and the working electrode [17]. The current flow between the modified working electrode (WE) and the counter electrode is recorded as the current response of the process [102].

The same process applies to the indolamines (serotonin) except for a slight difference in the redox electrode reaction. In this case, the hydroxyl group of the phenolic group loses its hydrogen during oxidation. Ditto for the hydrogen attached to the nitrogen of the indole ring. The converse occurs during reduction (Scheme 2).

## 6. Electrochemical Performance of Carbon-Based QD-Modified Electrodes

### 6.1. Electrochemical Performance in Dopamine Determination

Structurally, Dopamine (DA) can be considered a catecholamine due to the presence of catechol backbone in its chemical structure [68]. In solution, it assumes a cationic form which oftentimes makes its determination easy with electrodes bearing anionic entities at physiological pH [92]. This chemical nature further makes the electrochemical discrimination of DA achievable in the presence of popular interferents such as ascorbic acid (AA) and uric acid (UA), which are present in much higher concentrations in the human system and negatively charged at physiological pH [79]. Ordinarily, DA concentration in the human body ranges between 10–1000 nM [83]. It is a fact that ascorbic acid and uric acid (UA) have similar oxidation potential as that of DA but the field of nanotechnology have designed a lot of nanocomposites capable of selectively detecting DA in the presence of these interferents [79,92]. This occurs as a result of the modification of conventional gold, platinum, carbon paste, screen printed carbon, indium tin oxide (ITO) and glassy carbon electrodes with materials with anionic group for an electrostatic attraction or other forms of chemical interaction such as the π-π stacking [79]. In some cases, simultaneous determination of the three analytes has been done from ternary mixtures with various electrodes as a result of the separation of the oxidation potentials of the analytes by the electrode modifiers. Notably, those electrodes contain carbon nanotubes, metal and metal oxide nanoparticles [15,33], nafion [79], conducting polymers [35], molecularly imprinted polymers [120,121], chitosan [92], graphene oxide [34] and quantum dots [79,83,92]. The electrocatalytic effect brought to recent electrochemical sensors by the presence of graphene quantum dots and carbon quantum dots in the detection of DA is the crux of this section. Notably, there is a paucity of data on the use of graphene oxide nanoparticles for DA detection.

**Graphene quantum dots (GQDs)**, regardless of their synthetic route, are zero-dimensional nanomaterials with large surface area, high electrical conductivity, low toxicity, high solubility and good biocompatibility [98,122]. They mostly have sizes lower than 100 nm [98]. They are often synthesized from graphene—a material whose hydrophobic nature ordinarily makes it a pushover in electrochemical sensor design for biomolecules. Graphene quantum dots have also been synthesized from carbon black [98] and natural biological sources and applied for the fabrication of sensitive and selective electrochemical sensors. The possible π-π interaction and electrostatic interaction between DA and Graphene QDs (with sp^2^ hybridized carbon) is a theoretical basis for the selective DA detection by graphene QD containing composite in the presence of interferents [79].

A simple GQD-based sensor for the DA detection was reported by Zheng [81]. The simplicity of the electrode stems from the fabrication of GQDs from citric acid (CA) in a bottom-up fashion prior to the electrodeposition of the resultant GQDs on a polished glassy carbon electrode (GCE) in a cyclic voltammetry (CV) experiment. From chronocoulometry, it was discovered that the modified electrode tagged GQDs/GCE has an effective surface area that is ~7 times higher than that of bare GCE. Further characterization with electrochemical impedance spectroscopy (EIS) proved that the modification has reduced the resistance of the composite. These characterizations corroborated the smaller redox peak potential difference reported for the modified GCE (ΔEp = 25 mV) compared to that of bare GCE (ΔEp = 110 mV)—an indication that the modified electrode has a better electrocatalytic activity. An LOD of 50 nM at an LDR of 0.4–100 µM was obtained for this electrode in a differential pulse voltammetry (DPV) experiment. Selective DA detection was done in the presence of AA and UA with a DA-AA peak separation of 148 mV. The catalytic effect of the GQDs was largely due to the π-π interaction between DA and the QDs, as well as the electrostatic attraction between the anionic group on the QD and the cationic DA. Other interferents such as cysteine, tryptophan and inorganic salts did not interfere with the DA signal.

To design a sensor capable of discriminating AA and UA during DA detection is to assemble a platform with an anionic surface that attracts DA and repels AA and UA. In earlier literature, nafion, a polymeric material has provided such platform when incorporated into a composite. Therefore, a synergistic effect would be expected to occur if GQDs and nafion were found in a composite. Against this background, Jang et al. [98] fabricated an electrochemical sensor that contains these two materials and Au-nanoparticles (AuNPs) in a composite. The conductivity and the catalytic activity of the electrode were enhanced by the presence of the AuNPs. Essentially, the GQDs were prepared from carbon black in a single step process and made into a composite with AuNPs by dissolution in a solution of gold III chloride (HAuCl_4_). The composite was cast on GCE after nafion addition to obtain a working electrode (Au-GQDs/Nafion/GCE) with a better electrocatalytic ability than those containing AuNPs and GQDs alone (Au-GQDs/GCE). Using DPV, LOD of 0.84 µM within an LDR of 2–50 µM was obtained. Interestingly, the interference study was done with DA in the presence of AA, UA, glucose, NaCl, KCl and Na_2_SO_4_ using DPV. The electrode showed selectivity in the presence of the interferents without any change in the DPV current signal. As is evident from the composite making up the electrode, simultaneous determination of DA, UA and AA was not attempted.

Beyond the cheap source of precursor, the bottom-up synthesis of GQDs boasts of higher purity and solubility in aqueous medium [92]. Such GQDs and those prepared from other means have proven to have a better electrocatalytic ability, conductivity and chemical stability when doped with a heteroatom. This premise inspired the fabrication of nitrogen-doped GQDs (N-GQDs) and chitosan nanocomposite by Aoun et al. [92] for DA detection. The N-GQDs synthesized from glucose was made into a composite with chitosan prior to drop-casting on screen-printed carbon electrode (SPCE). The resultant working electrode (CS/N-GQDs/SPCE) was applied for the determination of DA in the presence of AA and UA using CV. The electrode proved selective and capable of simultaneous DA, AA and UA detection with DA-AA and DA-UA peak separations of 171 mV and 46 mV, respectively. This outcome was attributed to the anionic nature of chitosan at physiological pH. With DPV, the sensitivity of the electrode towards DA was reported as 418 µA mM^−1^ cm^−2^ while the LOD and LDR were 0.145 µM and 1–200 µM, respectively.

Pang and his group had previously, in 2016, presented an article on the design of GQDs-nafion nanocomposite modified GCE for DA determination in a publication that seems very much a replica of the study done by Jang et al. [98]. The differences between both working electrodes are the Au-Nanoparticle and the GQDs synthetic route. Through the top-down synthesis of GQDs from graphite via the hydrothermal route prior to casting on GCE and subsequent application of nafion, a working electrode coded GQDs-Nafion/GCE was fabricated. Again, the electrostatic and other chemical interactions between DA and GQDs coupled with the ionic discrimination ability of nafion at physiological pH led to the selective DA detection in the presence of AA and UA. Using the DPV technique for DA detection, LOD and LDR of 0.45 nM and 0.005–100 µM were obtained. The interference studies were done in the presence of UA, AA, glucose and inorganic salts such as NaCl, MgCl_2_, ZnSO_4_ and Ca(NO_3_)_2_.

Self-assembled monolayers (SAM) are known to possess good electrocatalytic ability, sensitivity and mass transport but poor stability [96]. To harness the gains of the SAM platforms and GQDs, Li et al. [96] designed a GQDs monolayer using ethylenediamine (eth) and GQDs, relying on the possible covalent interaction between the duo. The GQDs were synthesized from citric acid in a bottom-up approach and transformed to SAM-GQDs on GCE. The working electrode obtained (GQDs-eth/GCE) was applied for DA detection and an LOD of 0.115 µM was obtained within an LDR of 1–150 µM. The electrocatalytic ability of GQDs was ascribed to the performance of the electrode whose sensitivity was estimated as 1306 µA mM^−1^ cm^−2^. Simultaneous DPV determination of the DA, AA and UA determination done with DA-AA and DA-UA peak separation of 0.288 V and 0.194 V, respectively. The anionic carboxylic group on the GQDs which repels the anionic AA and UA was reported to have made the selective DA detection achievable. The presence of citric acid, mannose, fructose, lysine, cysteine, glucose, and inorganic ions did not affect the selectivity of the electrode.

Another N-doped GQD-based electrochemical sensor was fabricated by Hsu and Wu [84] for DA detection. In that study, tin II oxide SnO_2_, polyaniline (PANI) and N-GQDs were worked into a composite for the modification of GCE surface. PANI, a conducting polymer with poor mechanical strength was combined with SnO_2_ for improved mechanical and optoelectronic properties while trusting the electrocatalytic activity of N-GQDs. The N-GQDs, which were made from citric acid and urea, combined perfectly with PANI and SnO_2_ to give an electrode (SnO_2_/PANI/N-GQDs/GCE) with an LOD of 0.22 µM and an LDR of 0.5–200 µM in a DPV experiment. The selectivity of the electrode was evaluated in a ternary mixture of AA, DA and UA with DA-AA and DA-UA peak separations of 288 and 199 mV, respectively.

Silica materials such as N-[3(trimethoxysilyl)propyl^1^). They can also act as porous solid support for nanocomposites. Vinoth et al. [52] used TMSPED for the stabilization of Au nanocrystals (AuNCs) prior to the capping of the AuNCs with the GQDs. The interaction between the GQDs and the TMSPED was possible due to the covalent interaction between the amino group of the later and the carbonyl functional group on the GQDs. Notably, the GQDs were prepared from glucose via a simple bottom-up approach. An electrode tagged GQDs-TMSPED-AuNCs/GCE was obtained by casting the nanocomposite on bare GCE. This working electrode was used for the simultaneous determination of DA and EP with a peak difference of 412 mV in a DPV experiment. Through amperometry (AP), LOD of 5 nM and 0.01 µM were obtained for DA (LDR of 0.005–2.1 µM) and EP (LDR of 0.01–4 µM) detection, respectively. Sensitivities for DA and EP were estimated as 0.007 and 0.01 µA mM^−1^ cm^−2^, respectively. However, the determination of DA and EP in the presence of interferents was not attempted.

It is not alien to science that solving a problem comes close to creating another. The incorporation of noble metals into GQDs came with its pitfalls, such as the difficulty in forming the composite and the absence of stable 3D-architecture. This informed the diversion of attention to all carbon nanocomposite. Challenges associated with this arrangement such as large-sized particles and loss of electrocatalytic effect while crushing was addressed by Ruiyi et al. [97]. Histidine-functionalized GQDs (H-GQDs) and graphene micro aerogel (GMA) nanocomposite was developed for DA determination by Ruiyi and his group. The H-GQDs were prepared from histidine and citric acid. The working electrode formed (H-GQDs-GMA) had excellent electronic conductivity when compared to that of bare GCE, common graphene aerogel and the synthesized GMA. A much better LOD (0.29 nM) than the value reported by Jang et al. (0.84 µM) was obtained from this electrode when Au-NPs were incorporated into GQDs, with LDR as wide as 0.001–80 µM in a DPV experiment. The LOD reported in this study was equally lower than those estimated in every other GQD-based studies reviewed earlier. The selectivity of the sensor was confirmed by a difference of <2% in the DPV signal when DA was detected in the presence of inorganic ions. Interestingly, the electrode needed no extra solid support.

Multi-walled carbon nanotubes (MWCNTs) are known to be highly conducting types of nanomaterials with large surface area [113]. A whole lot of studies have reported their excellent electrocatalytic ability when combined with other nanomaterials such as metal and metal oxide nanoparticles. Arumugasamy and his group [66] designed GQDs and MWCNTs nanocomposite for GCE modification (Figure 12) to harness the possible synergy between these two materials. The GQDs in this study was made from glucose and dispersed in a solution of the acid-functionalized MWCNTs. The synergy between these two materials was evident in the stability of the resultant GCE modified electrode (GQDs/MWCNTs/GCE), electrocatalytic ability and sensitivity which manifested in obtaining an LOD as low as 95 nM at an LDR of 0.25–250 µM with DPV. The selectivity of the electrode was further validated in a solution of DA and interfering substances such as glucose, AA, KCl and glutamic acid using amperometry (AP). The DA current signal was not significantly affected by the presence of these interferents.

Beitollahi et al. [119] designed another very simple and selective GQD-based electrochemical sensor for DA detection in the presence of tyrosine (DA precursor). GQDs were prepared from citric acid by pyrolysis in a bottom-up fashion and cast on the graphite screen-printed electrode (SPE) to obtain a working electrode tagged GQDs/SPE. This electrode gave an LOD of 0.05 µM within an LDR of 0.1–1000 µM using DPV technique for DA determination while the LOD of tyrosine (tyr) was 0.5 µM at an LDR of 1–900 µM. The simultaneous determination of the two analytes was done with a peak separation of 435 mV. This result confirms the selectivity and sensitivity of this simple electrochemical sensor. Just like the electrode fabricated by Zheng et al. [81] where GQDs were directly cast on GCE, the simplicity of an electrode does not portend insensitivity.

**Carbon quantum dots (CQDs)**, since their discovery in 2004, have been considered a worthy alternative to the relatively toxic metallic semiconductor class of quantum dots [61]. This is also because CQDs are relatively smaller in size (2–10 nm), biocompatible, and cheaper to synthesize without losing the good fluorescence and electrical properties that QDs are known for [123,124]. As a result, they have found wide acceptance in the development of photovoltaic cells, bio-imaging, catalysis and electrochemical sensors design [80,125]. Just like the GQDs, they can be prepared from either the top-down or the bottom-up approach. The top-down approach involves synthesis from graphene and graphite oxide, while the bottom-up route has made use of natural green carbon sources like watermelon peel [56], grasses [59], pomelo peel [55] and sweet pepper [126]. A lot of CQDs have been synthesized from natural sources by carbonation and hydrothermal techniques. Other methods of CQDs synthesis involve laser ablation, solvothermal, electrochemical and microwave-assisted techniques [53]. Like GQDs, CQDs are also zero-dimensional quantum dots, but with sizes less than 10 nm [101]. They equally possess functional groups such as the carboxylate which makes their interaction with cationic species feasible [99]. Compared to other nanomaterials such as the CNTs, graphene oxide, metal and metal oxide nanoparticles, their application in the development of electrochemical sensors is not so pronounced.

Chen and his co-workers [100] designed a CQD-based electrode for DA determination in the presence of interfering UA and tryptophan (trp). The CQDs produced from the pyrolysis of citric acid was incorporated into β-cyclodextrin (CD) to form a composite that would inherit the conductivity and electrocatalytic activity of the CQDs, as well as the molecular selectivity of CD. This composite was electrochemically deposited on GCE to obtain a working electrode (CD-CQDs/GCE) that detected the three analytes in a ternary mixture (using DPV) with peak separations of 150 and 420 mV for DA-UA and UA-trp, respectively. LOD of 0.14 (LDR = 4–220 µM), 0.01 (LDR = 0.3–200 µM) and 0.16 µM (LDR = 5–270 µM) were obtained for DA, UA and trp, respectively. This stable electrode was applied for interference study in a solution containing the three analytes, glucose, L-cysteine, glutathione, citric acid, folic acid and metallic ions without a considerable difference in the initial current signal of the analytes. Compared to the values obtained from some GQD-based electrodes, the detection limit of this electrode is considerably low.

Chronoamperometry-controlled CQDs preparation from 1-propanol was carried out by Canevari et al. [51] for the simultaneous detection of DA and EP. The electrochemically synthesized CQDs were deposited on the surface of GCE. For comparison, SWCNTs and GO were used to modify GCE under the same condition for DA and EP determination in to identify the best electrode in terms of electrocatalytic activity. The GCE/CQDs electrode gave the best electrocatalytic activity towards DA and EP. Two variants of the GCE/CQDs electrode were fabricated to optimize the synthesis time of CQDs required for the best output. These were GCE/CQDs 4.5, which was made with 4.5 h of synthesis, and GCE/CQDs prepared for 8.5 h. The GCE/CQDs 4.5 gave a better current response to DA and EP. With DPV, LOD of 4.6 and 6.1 nM were obtained for DA and EP, respectively using GCE/CQDs 4.5 electrode. Using the GCE/CQDs 8.5 electrode, slightly higher LODs for DA (6 nM) and EP (6.5 nM) were obtained. The LDR estimated for DA and EP using GCE/CQDs 4.5 was 0.05–2 µM apiece. The sp^2^ hybridized carbons, the small size (3 nm) of the carbon dots and the presence of the carboxylic group which foster interaction with the biomolecules were the reasons given for the simultaneous detection of DA and EP. These analytes were selectively detected with the GCE/CQDs electrodes in the presence of AA and UA. The LOD obtained from this work is much lower than the value reported by Chen et al. [100] and that of some GQD-based sensors.

Algarra and his team developed a CQD-based electrochemical sensor similar to that of Canevari et al. The differences being that Algarra et al. [64] synthesized the CQDs used for electrode modification from pencil graphite via a top-down approach. Beyond this, a comparative study with other carbon-based materials was not done. The synthesized CQDs were deposited on the GCE to obtain a working electrode (GCE/CQDs) and applied for the detection of UA and DA. LOD of 2.7 and 1.3 µM were obtained for DA and UA, respectively, using the linear sweep voltammetry within respective LDR of 0.19–11.81 µM and 0.21–31.39 µM. Interference studies were not conducted in this study because the detection of UA was done independently, possibly due to the biological significance of UA.

Chitosan (CS) has been used alongside CNTs in the design of electrochemical sensors with good sensitivity. This was replicated by Huang et al. [80] in the development of CQDs and CS (CQDs-CS) composite modified GCE (CQDs-CS/GCE) for DA detection (Figure 13). The CQDs were synthesized from polyethylene glycol and glucose using a microwave-assisted technique. Though chronocoulometry, DA diffusion coefficient was reported to be 3.68 × 10^−6^ cm^2^ s^−1^, which implies fast electron transfer across the developed material. Selective DA detection in the presence of AA and UA was successfully done at physiological pH because AA and UA exist as anions at that pH (7), while the cationic DA at that pH exhibit an electrostatic attraction for the CQDs-CS composite with anionic groups. Even in the presence of interferents such as glucose, L-cysteine, lysine, NE and notable cations and anions, the current response of the electrode to DA was not significantly affected. LOD of 11.2 nM at an LDR of 0.1–30 µM was obtained using DPV for DA detection. This LOD is satisfactory for DA detection, and at the same time much lower than the values reported for some electrochemical sensors based on GQDs.

Aggregation of graphene molecules due to weak intermolecular force of attraction such as van Der Waal force necessitates the functionalization of graphene with molecules such as ionic liquids (IL) and polymeric materials capable of enhancing its dispersibility and solubility in common solvents. The incorporation of IL functionalized graphene into CQDs synthesized from graphene oxide presented a sensitive, stable and selective platform for DA detection. The working electrode developed in this study tagged IL-graphene/CQDs/GCE was applied for the determination of DA in the presence of AA, UA, lysine, cysteine, glucose and other ions with negligible DA current signal response. Zhuang et al. [76] further stressed that using this electrode, DA can be detected up to a concentration of 30 nM via DPV at a linear range of 0.1–600 µM. Zhuang and his team, just like some other research groups, ascribed the discrimination of AA in the DA detection to the presence of the anionic carboxyl groups on the CQDs. The LOD obtained is still relatively low compared to that of the GQD-based electrode designed by Arumugasamy and his group [113].

GQDs exhibit improved electrocatalytic activity, conductivity, tunable fluorescence and chemical stability when doped with heteroatoms such as nitrogen. According to Jiang and his group, doping CQDs with nitrogen induces produces a matrix capable of inducing charge delocalization. Building on this premise, Jiang et al. [83] fabricated nitrogen-doped CQDs from collagen and ammonia (NH_3_) via hydrothermal reaction and subsequently drop-casted on bare GCE in nafion solution to obtain a working electrode coded NCQDs-nafion/GCE. This electrode gave an LOD of 1 nM at an LDR of 0–1 mM using DPV. The simultaneous DA detection in the presence of AA and UA was achieved with peak separations of 760 mV and 340 mV for DA-UA and AA-UA, respectively. The LOD reported for this electrode is much lower than the value obtained by Hsu and Wu [84] (0.22 µM) and Ben Aoun [92] (0.145 µM) in articles where N-GQDs were used. Coincidentally, another research group belonging to Jiang et al. [83], in the same year—2015—designed another nitrogen-doped CQDs for DA determination. Essentially, a single material diethanolamine (DEA) served as the C and N source for the synthesis of microwave-assisted N-CQDs subsequently used for GCE modification.

The microwave means of synthesis proved very fast and economical because the synthesis lasted 10 min without the use of solvent. A detection limit as low as 1.2 nM within a linear range of 0.05–800 µM was obtained for DA determination in a DPV experiment. The carboxylic acids and amino functional groups on N-CQDs were attributed to the electrode’s electrocatalytic activity. The selectivity of the electrode was proven through the detection of DA in the presence of AA with peak separation as high as 580 mV. The LOD obtained in this study is close to the value reported by Jiang et al. [62] (1 nM) using another N-CQD-modified electrode.

In one of the early publications on the use of CQD-based sensor for DA determination, Li et al. [94] designed an electrochemical sensor using CQDs synthesized from glucose in a solution of glycol. The CQDs obtained was cast on the surface of GCE to obtain the working electrode (CQDs/GCE) for DA determination. This simple electrode was applied for DA detection and an LOD of 26 nM was obtained at an LDR of 0.15–150 µM using DPV. This LOD is much higher than the value reported by Jiang et al. where N-CQDs/GCE was used but some orders of magnitude lower than that of Algarra et al. (2.7 µM), where the working electrode was also CQDs/GCE. The major difference between the electrodes fabricated by Li et al. and Algarra et al. is the preparation of the CQDs. The latter obtained the CQDs from pencil graphite. Li et al. [94] further demonstrated the selectivity of this electrode in a ternary mixture of DA, AA and UA where DA was selectively determined. The cation exchange ability of CQDs was ascribed to this selectivity, this time at a pH of 6.

The conductivity, large surface area and biocompatibility of Au nanoparticles coupled with the fascinating electrochemical properties of CQDs inspired Huang et al. [99] to develop an AuNP, CQDs and chitosan (CS) nanocomposite for GCE modification. The CQDs were produced from sucrose and oil acid via a bottom-up route. The resultant electrode (AuNP/CQDs-CS/GCE) was characterized and applied for DA detection using DPV technique. LOD of 1.0 nM within an LDR of 0.1–30 µM was obtained and the electrode proved very selective for DA in the presence of primary (AA and UA) and secondary interferents like cysteine, glucose, lysine, citric acid, aspartic acid and common physiological ions. Clearly, the synergy between AuNPs and the CQDs have given a discriminatory platform for the interferents and at the same time impacted on the sensitivity of the electrode.

Copper I oxide (Cu_2_O) nanoparticles are metallic oxide nanoparticles with good stability and high catalytic activity. These decent properties coupled with conductivity, biocompatibility and stability of CQDs informed the synthesis of the Cu_2_O-CQDs/Nafion composite for modification of GCE by Huang et al. [82]. The inclusion of Nafion was largely to boost the specificity of the resultant electrode (Cu_2_O-CQDs/Nafion/GCE) for DA in the presence of AA and UA, due to its cation exchange ability. Yet again, the CQDs were produced from sucrose and oil acid via carbonization. This working electrode was applied for selective DA detection in the presence of AA and UA with good separations without significant change in the DA current signal. This same observation persisted in the presence of lysine, glucose, cysteine and other ions of physiological importance. The LOD obtained in this study (1.1 nM) was very close to the value reported by Huang et al. the previous year using AuNP/CQDs-CS/GCE electrode but this happened over a different LDR (0.05–45 µM).

CQD-based electrode for a comparative study was designed by Devi et al. [101] for DA determination. CQDs synthesized from graphite through electrochemical means were used for the modification of screen-printed carbon electrode (SPCE) and GCE and applied for DA detection to ascertain the electrocatalytic ability of the CQDs. The CQD-modified electrodes performed much better than the bare electrodes judging by their respective DA current responses. CQDs/SPCE was used for further electrochemical studies and an LOD of 99 nM was obtained at an LDR of 1–7 µM using cyclic voltammetry. The electrostatic attraction of DA by the carboxylate and hydroxyl ions attached to the CQDs was ascribed to their electrocatalytic ability. For the electrode to offer a selective DA determination in the presence of AA and UA, the working electrode was modified with nafion to benefit from its cation exchange ability. Compared to a similar electrode prepared by Beitollahi et al. (GQDs/GCE) in the same year (2018), where a much higher LOD was obtained (0.5 µM); this electrode has a satisfactorily low LOD.

These studies, as highlighted in this section and summarized in Table 1, represent efforts geared towards the electrochemical determination of DA using GQDs and CQDs from 2010 to date. As seen in Table 1, the lowest of the detection limits for DA determination using the two carbon-based quantum dots came from GQDs (0.29 nM). This was obtained from histidine functionalized GQDs. This is not so surprising because, apart from the customary carboxylate and hydroxyl functional group on the GQD, the presence of histidine supported the introduction of amino functional groups, thus increasing the number of binding sites available for DA and the H-GQDs. For CQDs, the lowest LOD (1.0 nM) for the CQDs was jointly recorded by Huang et al. [99] using AuNP/CQDs-CS/GCE electrode and Jiang et al. [83] with NCQDs-nafion/GCE. The precursors for the CQDs were sucrose and collagen, respectively. Comparing this result with the low LOD obtained from other CQD-based sensors, with CQDs from other carbohydrate sources, it can be inferred that the source of the CQDs impacts the electrocatalytic activity of the resultant quantum dots. Looking at the publications coming from both ends (GQDs and CQDs), the number of electrodes with LOD < 10 nM are found to be CQD-based. Incidentally, the CQDs in these electrodes were fabricated from carbohydrate sources. Although the choice of these starting materials for CQDs manufacture was primarily based on availability, the relatively low detection limit obtained makes them a worthy component of future CQDs. More importantly, the two classes of QDs have succeeded in giving rise to reliable electrodes for DA determination judging from these LODs and the wide linear dynamic ranges (LDRs) estimated.

Selective determination of DA in the presence of primary interferents such as AA and UA is almost impossible without adequate modification of the conventional electrodes. Achieving the electrochemical discrimination of these molecules requires a good understanding of their behaviour at physiological pH. Thankfully, such pH levels (6–7.4) were used in the publications under review (Table 1). It is a fact that AA and UA assume an anionic form in physiological pH while DA maintains a cationic nature (Figure 14). To attract the analyte of interest, the working electrode must be capable of repelling interferents, and this type of electrode often has cation exchange material incorporated into its modifying composite. Nafion and chitosan are notable materials that have earned a great reputation as discriminatory agents in selective electrochemical DA determination due to their anionic nature. Several publications within the years under review did not attempt to establish the peak difference between the oxidation potential of DA and the interferents because the selective determination of DA was investigated. Where such peak differences were established, the possibility of simultaneous DA detection was tested and then, establishing the peak difference became a necessity. This explains why few peak differences appeared in Table 2, and the absence of nafion or chitosan in the composition of some electrodes that showed the peak differences. The peak difference is an analytical figure of merit that shows the feasibility of simultaneous determination of multiple analytes. Ion discriminating materials might not be necessary for this to be achieved, the basic criterion is an electrode that can detect such analytes at oxidation potentials wide enough to establish peak difference(s). As seen in Table 2, in instances where nafion and chitosan were relied upon to boost the peak separations, satisfactory results were obtained; and where such materials were not incorporated, the simultaneous DA detection in the presence of interferents was equally achieved. Very few data are available for the simultaneous electrochemical DA detection in the presence of AA and UA using carbon-based QDs. Fewer could be found for the simultaneous DA and tryptophan (or tyrosine) determination. These might just be some other areas of focus for the next set of novel carbon-based QDs DA electrochemical sensors. 

### 6.2. Electrochemical Performance in Epinephrine (EP) Determination

Greater attention has always been given to DA compared to other neurotransmitters. This could be due to its physiological significance and the implication of its imbalance in the nervous system. Succinctly put, other neurotransmitters lack serious attention in terms of electrochemical detection compared to DA. This was the case with carbon nanotubes-based sensors and it is no different with quantum dots-based electrochemical sensors. Very few publications are available on the electrochemical determination of EP with GQD- and CQD-based sensors. Epinephrine, otherwise called adrenaline, is an excitatory neurotransmitter that doubles as a hormone [93]. EP exists as a cationic biomolecule in extracellular fluids and the depletion of its concentration could be linked to a chronic psychological disorder such as Parkinson’s disease [86]. Specifically, in the human body, EP exists at a control level of 0.09–0.69 ng/mL [127]. Activities that are capable of causing a persistent surge in the human EP body balance can lead to chronic diseases like high blood pressure [128]. Consequently, its determination in body fluids gives a clue in clinical diagnosis for diseases associated with EP level in extracellular fluid. A lot of research has gone into EP determination using nanomaterials with good sensitivity [129,130,131]. This section focuses on the use of carbon-based quantum dots for EP determination.

**Graphene quantum dot (GQD)**-based electrochemical sensors have been used for the determination of EP individually and in the presence of interferents with reliable sensitivity. One such GQD-based electrochemical sensor was designed by Tashkhourian et al. [86]. Essentially, the electrode was a carbon paste electrode modified with GQDs. The stability of the electrode was boosted by the introduction of chitosan as a binder, relying on the electrostatic attraction between the anionic carboxylate ions on the GQDs and the protonated amino group in the linear chain of chitosan under low pH. This property of chitosan, high surface area and conductivity further made it the chosen material for making the composite. The GQDs were prepared from citric acid and mixed with graphite powder and chitosan to obtain the working electrode (GQDs/CS/CPE) (Figure 15). This electrode was applied for EP determination using chronoamperometry and an LOD of 0.3 nM was recorded within an LDR of 0.36–380 µM. The selectivity was also established with the detection of EP in the presence of both organic and inorganic interferents. This electrode compares favourably with another GQD-based electrochemical sensor developed by Vinoth et al. [52], as discussed in the previous section. With GQDs-TMSPED-AuNCs/GCE, Vinoth et al. reported an LOD of 10 nM for EP determination, which is some orders of magnitude higher than 0.3 nM.

In the same vein, Baluta et al. [114] fabricated a biosensor based on GQDs made from the pyrolysis of citric acid as well. The uniqueness of this electrode is in the introduction of an enzyme (laccase) to obtain an electrochemical sensor with improved sensitivity and selectivity for EP. The enzyme was immobilized on the GQD-modified GCE with the aid of glutaraldehyde which serves as a crosslinking agent, thus enabling the covalent attachment of the enzyme to the electrode.

The GQDs serve as the support upon which the laccase which assumed the position of a catalyst for the reduction and oxidation of EP is placed. The possession of the Cu I and II redox-active centre made the electrocatalytic action of laccase feasible. The presence of this redox couple on laccase was validated with CV. This catalytic effect of the enzyme is evident in the determination of EP in a CV experiment using the GCE/GQDs/Lac electrode with good sensitivity. LOD and LDR of 83 nM and 1–120 µM, respectively were obtained with this electrode. This LOD is much higher than the value reported by Tashkhourian et al. [86], where chitosan was applied on GQDs support for modifying CPE. The selectivity of the electrode was established with the determination of EP in the presence of AA, UA, cysteine, tryptophan and glutathione with negligible alteration in the peak current.

**Carbon quantum dots (CQDs)** have equally been used for the fabrication of electrochemical sensors for EP determination within the years under review. Compared to the CQD-based electrode for DA determination, these type of electrodes are relatively few. A good example is an ultrasensitive EP imprinted electrode designed by Yola and Atar [116] for the determination of EP. The field of molecular imprinting has achieved a lot of success in the design of electrochemical sensors because of their native selectivity and specificity. This and the special electrocatalytic activity of nitrogen-doped graphene QDs (N-CQDs) would have been the motivation for Yola and his group to develop imprinted graphitic carbon nitride and N-GQDs composite for EP determination. The graphitic carbon nitride was prepared from melamine while the N-GQDs was made from citric acid and urea. The resultant composite was cast onto the surface of the GCE prior to the electrodeposition of pyrrole and EP on the working electrode to obtain an electrode tagged MIP/g-C_3_N_4_/NCQDs/GCE. The non-imprinted analogue was prepared as a control. As expected, the imprinted electrode showed a better electrocatalytic activity toward EP compared to the non-imprinted analogue. This was evident from the much better current output recorded for the imprinted electrode. Using DPV, an LOD (0.0003 nM) much lower than the one reported for some other imprinted electrodes was reported for this electrode within an LDR of 0.001–1 nM. This electrode was successfully used for selective EP determination in the presence of DA, AA, UA, NE, tyr and trp. Although these interferents were detected at about the same oxidation potential as EP, the outstanding current response of EP made the selectivity test achievable.

A very simple CQD-modified carbon paste electrode (CPE) was developed by Shankar et al. [93] for simultaneous EP, AA and SE determination. The CQDs were prepared from styrene and applied for the modification of CPE. The working electrode obtained (CQDs/CPE) was used for EP determination in a chronoamperometry experiment. LOD of 6 nM was obtained at an LDR of 0.02–20 µM. This electrode was further used for the individual determination of AA and SE with LODs of 60 nM and 4 nM, respectively, recorded at a respective LDR of 0.1–10 and 0.01–8 µM. With CV, simultaneous determination of a ternary mixture of these analytes was done with AA-EP and EP-SE peak separations of 197 and 179 mV, respectively. The interference of the other two analytes with EP was investigated with DPV. The current response of the electrode to EP signal was not significantly affected and this proves the selectivity of the electrode. The strength of this electrode lies in the anionic nature of the CQDs which attracts the cationic EP.

### 6.3. Electrochemical Performance in Serotonin (SE) Determination

Serotonin is also a very common and essential neurotransmitter. SE is largely made in the central nervous system and the intestine [95]. It is one of the inhibitory neurotransmitters and is therefore important for the regulation of the calm physiological human behaviour such as learning, sleeping and mood [132]. Anxiety is one of the manifestations of an unusually low amount of SE in the human system, while a condition termed serotonin syndrome can be attached to a rise in its level [133]. The symptoms of serotonin syndrome are high body temperature, diarrhoea, increased reflexes, and agitation. A lot of studies are available for the determination of SE in biological fluids and pharmaceutical samples using different types of nanomaterials [15,134,135], but very few can be found on the determination of SE using both CQDs and GQDs. One of the very few publications on carbon-based quantum dots for EP determination was presented by [95]. The stability, conductivity and the large surface area of boron nitride and the electrocatalytic activity of GQDs motivated Yola and Atar to design a hexagonal boron nitride and GQD-modified GCE. The GQDs were prepared from graphite in a top-down approach. The resultant electrode (h-BN/GQDs/GCE) was further subjected to electrodeposition of EP to form an imprinted version of the working electrode (MIP/h-BN/GQDs/GCE). The superior electrocatalytic activity of these electrodes compared to that of the non-imprinted analogue manifested in higher CV current response. This SE imprinted electrode gave a detection limit of 0.0002 nM at an LDR of 0.001–10 nM. The LOD recorded here is much lower than the values reported in literature for SE detection where other nanomaterials have been used for the fabrication of the electrode [15,134,135]. To the best of our knowledge, this is the only study solely on the electrochemical determination of SE with QDs. A closer study was the synthesis of room temperature ionic liquid (RT-IL) and GQD nanocomposites for the modification of carbon paste electrode for Levodopa determination in the presence of SE [77]. SE was only considered an interferent in the study because SE neurons are capable of decarboxylating levodopa to DA, thus affecting the SE systems adversely [136]. Similarly, Shankar et al. used a CQD-modified CPE originally designed for EP detection for simultaneous EP, SE and AA detection to ascertain the selectivity of the electrode, but an LOD of 4 nM at an LDR of 0.01–8 µM for SE in a DPV experiment is noteworthy.

### 6.4. Electrochemical Performance in Norepinephrine (NE) Determination

Norepinephrine (NE), also known as noradrenaline, is an excitatory neurotransmitter produced in the central nervous system from DA in a synthetic pathway similar to that of DA—except for the transformation of DA to NE by dopamine β-hydroxylase [137]. After the reuptake of NE, it can be broken down further by monoamine oxidase (MAO). NE can equally double as a stress hormone. NE is popularly referred to as the ‘flight or fight’ NT because of its ability to activate the sympathetic nervous system in moments of danger [19]. It is responsible for the regulation of heart rate, blood pressure, body metabolic rate and temperature and memory [106,138]. Specifically, anxiety-related disorders such as obsessive-compulsive disorder (OCD), post-traumatic stress disorder (PTSD) and panic disorder have been treated with MAO [138]. Like every other monoamine NTs, the health implication of the imbalance of NE such as mental disorders and heart failure [139] has triggered the development of several electrochemical and fluorescence sensors for its detection. Unfortunately, just like other SE and EP, very few studies are available for the detection of carbon-based quantum dots of NE. Specifically, the few available ones are GQD-based.

**Graphene quantum dot (GQD)**- based NE detection was performed by Fajardo and his co-workers [106] by using GQDs synthesized from citric acid via pyrolysis in a bottom-up approach. Yet again, the large surface area, biocompatibility and low toxicity of the quantum dots coupled with the excellent conductivity of Au nanoparticles inspired the production of the GQDs and AuNPs composite modified GCE for NE detection. This nanocomposite was previously prepared by Jang et al. [98] in 2019 in a comparative study with Au-GQDs/Nafion/GCE for DA detection. The major difference being that Jang and his group prepared the GQDs with carbon black. Fajardo et al. [106], after confirming the superiority of the GQDs-AuNPs/GCE electrode over the GQDs/GCE, proceeded to carry out electrochemical detection of NE using square wave stripping voltammetry (SWSV) at a physiological pH of 7. LOD of 0.15 µM at an LDR of 0.5–7.5 µM were obtained using this electrode in an SWSV experiment. The selectivity of the electrode was proven by the selective NE detection in the presence of AA, UA and other pharmaceutical samples which would be expected to interfere with the NE signal. Specifically, AA and UA showed signals at different oxidation potential as NE with peak separations wide enough for selective NE determination. Detection of NE in injection samples and rat brain tissue using the standard addition technique gave recoveries that confirmed the suitability of the electrode for real-life NE detection.

The apparent importance of NE and acetylcholine (ACh) for the well-being of the nervous system has been established. More importantly, the duo plays a vital role in Parkinson’s disease and other neurological disorders. This reality prompted Jahani [111] to design an electrochemical sensor for the detection of NE and ACh. Since this review is basically on the detection of monoamine NTs with carbon-based QDs, the spotlight would be largely on the efforts of Mohammadzadeh and his group in NE detection. With the knowledge of the ionic conductivity of ionic liquids and the boost in the electrocatalytic activity of composites when IL is introduced into the matrix, GQDs-Carbon paste-IL composite was made for NE and ACh detection. The resultant electrode, GQDs/IL/CPE, was used for the detection of NE in a DPV experiment. LOD and LDR of 0.06 µM and 0.2–400 µM were obtained. This LOD is extremely low compared to the value reported by Fajardo et al. [106]. The electrode was further applied for the simultaneous NE and ACh detection with peak difference of 309 mV, suggesting possible simultaneous NE and ACh detection without the overlapping of signals. Real sample analysis of the analytes in urine and pharmaceutical samples of the analytes was achieved using the standard addition technique as well. The high percentage recovery of the analytes was a strong indication that the electrode can be applied in real clinical samples. Notably, the quantum dots were made from the pyrolysis of citric acid.

Over the last ten years, compared to SE and NE, EP has witnessed greater attention as an analyte for detection with carbon-based QDs. The available literature shows that the greater percentage of such carbon-based QDs were synthesized from citric acid (Table 3). This in part limits the definition of GQDs as QDs made from graphene/graphene oxide. Using the LOD, the greatest LOD for EP detection came from a MIP-based sensor. It would not be completely right to ascribe the performance of the electrode to the electrocatalytic activity of the CQDs only. This is because the selectivity and specificity supplied by the MIP is too important to be ignored. This reality and the paucity of data makes it difficult to outrightly conclude that the CQDs have performed better for EP detection than the GQDs. This equally applies to the determination of SE. To the best of our knowledge, there was no data on the determination of NE with CQD-based electrochemical sensor.

## 7. Carbon-Based QDs and Monoamine Neurotransmitters: Now and Future Perspective

As seen in sensors based on other types of carbon-based nanomaterials like CNTs, graphene oxide and other nanoparticles, dopamine has yet again been given greater attention in the development of carbon-based quantum dot-enhanced electrochemical sensors. This can be ascribed to its physiological importance, which possibly stems from its dual inhibitory and excitatory function in the nervous system. Based on the literature available within the years under review, the LODs obtained still leave more to be desired, because much lower LODs have been reported in some other studies on DA detection [140,141]. The very few articles found in EP, NE and SE determination gave very low LOD for the detection of the respective neurotransmitters. It could be observed that the determination of these analytes using molecular imprinting yielded electrodes with the capacity to detect these analytes to the picomolar level. This is largely due to the specificity and the selectivity of imprinted polymers which raises the electron transfer kinetics. Since the path of the molecules in solution has been defined through molecular imprinting, the influx of electron towards the receptor (modified electrode) becomes much easier. These results are an indication that further work with an outlook of CQDs/GQD-modified with various imprinted polymers would most likely give electrodes with much lower detection limits. A matrix with improved conductivity emerges when heteroatoms are applied for doping carbon-based quantum dots. This is a result of the enhanced electron delocalization that emerges from such a matrix. This conductivity has been proven in the available literature discussed in this review, but such studies are very few. Future monoamine NT electrochemical sensors with nitrogen- and other heteroatom-doped carbon-based quantum dots incorporated into a composite with other highly catalytic and conducting nanomaterials might offer the desired sensitivity.

To the best of our knowledge, and within the time frame considered for this review, there was only one study on the determination of monoamine NTs using CQDs and GQDs combined with enzymes. Enzyme laccase was only used for the determination of EP with LOD as low as 83 nM reported. This is an indication that enzymes can favourably be combined with carbon-based quantum dots for the detection of neurotransmitters. Similarly, the only available data for the incorporation of amino acids into carbon-based QDs was done with histidine. The resultant electrode gave an LOD of about 0.29 nM, which is a reasonably low LOD for DA detection. Although the sensitivity might not be down to the amino acid alone, investigating this by modifying carbon-based QDs with amino acids in future electrochemical sensors might give ultrasensitive electrodes. On a general note, GQDs and CQDs have both been very rewarding in terms of sensitivity and selectivity based on the fact that both selective and sensitive electrodes have been made from the duo within the years under review with low LOD. However, most of the precursors in each case have come from carbohydrates and citric acid, which adds a bit to the cost of production of the sensor. Electrochemical techniques compared to other means of detection of neurotransmitters have a huge reputation for cost efficiency. Therefore, it is important to look more into the possibility of preparing future carbon-based quantum dots for monoamine neurotransmitter detection using biological waste with a huge amount of carbon.

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
