# Peer review of "Carbon-Based Quantum Dots for Electrochemical Detection of Monoamine Neurotransmitters—Review"

_biosensors, 2020, doi:10.3390/bios10110162_

Round 1
Reviewer 1 Report
In this manuscript, electrochemical sensing applying carbon based quantum dots sensors for Monoamine Neurotransmitters detection are reviewed. In my opinion this work is really interested, well described and fully collected. Nevertheless, there are few issues that need to be addressed ahead of publication. Therefore, I believe that this manuscript should be published in Biosensors after minor revision.
- Keywords are missing.
- English is quite poor but completely understandable. In my opinion, the work can be linguistically improved.
- The quality of the figures is very poor. It has to be improved.
- Verse nr 138 - surname Sajid should be written using capital letter.
- Figures in the text are taken from the original works. In my opinion, authors should prepare their own versions of these figures based on them (at least partially).
- It is generally known that, the choice of an appropriate supporting electrolyte and pH conditions is a significant step in electrochemical measurements as the reaction medium has an appreciable effect on the electrode mechanism. In my opinion in the tables presented in the manuscript should be added the type of used supporting electrolyte for each item.
- Furthermore, the tables should contain the slopes (sensitivities) and tested interfering agents for each item.
Reviewer 2 Report
The review “Carbon Based Quantum Dots for Electrochemical Detection of Monoamine Neurotransmitters – A Mini Review” by Saheed E. Elugoke, Abolanle S. Adekunle, Omolola E. Fayemi, Bhekie B. Mamba, El-Sayed M. Sherif, Eno E. Ebenso describes a highly interesting topic. The title is not clear in respect to “Mini Review”since the length is appropriate for an usual review.
Electrochemical method is a straightforward strategy to characterize monoamine neurotransmitters since excitation of nerve cells is also an electrical phenomenon. An innovative idea for chemical sensing are quantum dots. Their size leads to specific properties in respect to their optical and electronic properties. This paper deals with carbon quantum dots which can be synthesized in different pathways. Their advantageous features are superior in many aspects to metallic or semiconducting analogues, since they are not toxic.
Dopamine determination, Epinephrine determination, Serotonin and Norepinephrine were described. The most papers are published on dopamine. The detection of these neurotransmitters follows the scheme to form a quinone.
The review is a very comprehensive information for all researchers who would like to get insight in this field. Generally, a problem arises in such a type of review between the extent of the paper and profound details. In this case many papers were cited and not only some key publications.
Reviewer 3 Report
In this review the authors summarized the recent advances and remarkable studies of carbon based quantum dots including carbon and graphene quantum dots (CQDs and GQDs), in terms of their structures, preparations, properties and electrochemical biosensor applications for neurotransmitters. Moreover, they discussed the challenges facing CQD and GQD-based electrochemical sensors and their future perspectives.
Thus, the manuscript of this review is timely to the field and tightly focused. The topic is strongly interesting and suitable for Biosensors.
There are three points that could be included/modified to further improve the quality of the manuscript.
- In Peak difference of table 1, values or crosshairs should be aligned to the center.
- Figure 8 is blurry. Please change the figure to a clear one.
- The authors could make a brief table so that the readers can clearly understand the similarities and differences between CQD and GQD.
Reviewer 4 Report
Please see the attached file.

Round 2
Reviewer 4 Report
No further query.